# Enhancing Heart Disease Diagnosis Using ECG Signal Reconstruction and Deep Transfer Learning Classification with Optional SVM Integration

**DOI:** 10.3390/diagnostics15121501

**Published:** 2025-06-13

**Authors:** Mostafa Ahmad, Ali Ahmed, Hasan Hashim, Mohammed Farsi, Nader Mahmoud

**Affiliations:** 1Computer Science Department, Faculty of Computers and Information, Menoufia University, Shibin el Kom, Menofia Governorate 6131567, Egyptnader.mahmoud@ci.menofia.edu.eg (N.M.); 2Department of Computer Science and AI, College of Computing and Information Technology, University of Bisha, Bisha 67714, Saudi Arabia; 3Information Technology Department, Faculty of Computers and Information, Menoufia University, Shibin el Kom, Menofia Governorate 6131567, Egypt; 4Department of Information Systems, College of Computer Science and Engineering, Taibah University, Taibah 42353, Saudi Arabia; hhashim@taibahu.edu.sa (H.H.); mafarsi@taibahu.edu.sa (M.F.); 5Cybersecurity Department, Engineering and Information Technology College, Buraydah Private Colleges, Buraydah 51411, Saudi Arabia

**Keywords:** medical images, signal image processing and classification, heart failure diagnosis, DL models

## Abstract

**Background/Objectives:** Accurate and efficient diagnosis of heart disease through electrocardiogram (ECG) analysis remains a critical challenge in clinical practice due to noise interference, morphological variability, and the complexity of overlapping cardiac signals. **Methods**: This study presents a comprehensive deep learning (DL) framework that integrates advanced ECG signal segmentation with transfer learning-based classification, aimed at improving diagnostic performance. The proposed ECG segmentation algorithm introduces a distinct and original approach compared to prior research by integrating adaptive preprocessing, histogram-based lead separation, and robust point-tracking techniques into a unified framework. While most earlier studies have addressed ECG image processing using basic filtering, fixed-region cropping, or template matching, our method uniquely focuses on automated and precise reconstruction of individual ECG leads from noisy and overlapping multi-lead images—a challenge often overlooked in previous work. This innovative segmentation strategy significantly enhances signal clarity and enables the extraction of richer and more localized features, boosting the performance of DL classifiers. The dataset utilized in this work of 12 lead-based standard ECG images consists of four primary classes. **Results**: Experiments conducted using various DL models—such as VGG16, VGG19, ResNet50, InceptionNetV2, and GoogleNet—reveal that segmentation notably enhances model performance in terms of recall, precision, and F1 score. The hybrid VGG19 + SVM model achieved 98.01% and 100% accuracy in multi-class classification, along with average accuracies of 99% and 97.95% in binary classification tasks using the original and reconstructed datasets, respectively. **Conclusions**: The results highlight the superiority of deep, feature-rich models in handling reconstructed ECG signals and confirm the value of segmentation as a critical preprocessing step. These findings underscore the importance of effective ECG segmentation in DL applications for automated heart disease diagnosis, offering a more reliable and accurate solution.

## 1. Introduction

Heart failure remains a significant global health challenge, affecting millions of individuals and imposing a heavy burden on healthcare systems [1]. Early and accurate diagnosis is critical for effective treatment and improved patient outcomes. Traditionally, the diagnosis of heart failure has relied heavily on electrocardiogram (ECG) analysis, often interpreted manually by clinicians. While ECGs are a vital tool for detecting cardiac abnormalities, traditional methods face limitations such as subjectivity, reliance on expert interpretation, and susceptibility to human error. These challenges can lead to delayed or missed diagnosis, particularly in cases with subtle or complex abnormalities [2].

In recent years, advancements in artificial intelligence and DL have revolutionized the field of medical diagnostics. By leveraging large datasets and powerful computational models, DL has demonstrated exceptional potential in analyzing complex medical data, including ECG images, with higher accuracy and efficiency than traditional methods. The application of DL in diagnosing heart failure offers a paradigm shift, enabling automated detection, feature extraction, and predictive analytics, all of which are critical in enhancing diagnostic precision [3].

Segmenting ECG signals accurately is a critical step for diagnosing heart failure, but it poses significant challenges. ECG signals are inherently complex, exhibiting variability due to patient-specific factors, noise, and artifact distortions. Additionally, heart failure may manifest subtly or overlap with other conditions, making clear segmentation difficult [4,5,6]. To address these issues, advanced signal-processing techniques and DL models have been proposed. DL models, particularly convolutional neural networks and recurrent neural networks, excel in automatic feature extraction and accurate segmentation. Furthermore, denoising techniques and diverse datasets improve reliability and performance.

This paper explores the challenges of diagnosing heart failure through traditional ECG image analysis and emphasizes how effective segmentation techniques and DL models are helping to address these issues. Furthermore, it emphasizes the broader applications of DL in the diagnosis and management of various medical conditions, highlighting its growing importance in modern healthcare.

The key contribution of this paper is demonstrating how ECG signals, even when overlapping or noisy, can be effectively used for accurate ECG image segmentation and heart failure classification by leveraging DL and Machine Learning approaches. Specifically, this paper proposes the following:A comprehensive framework for ECG signal analysis that integrates image segmentation and transfer learning classification, addressing challenges such as noise interference and waveform variability.An effective segmentation approach that enhances the extraction of ECG signals, improving the clarity of the signals for better classification and diagnosis.Rigorous evaluations of several transfer learning models—such as VGG19, VGG16, InceptionNetV2, ResNet50, GoogleNet, and others—for both binary and multi-class classification tasks, all of which demonstrated high accuracy rates.A hybrid model combining VGG19 with SVM, which achieved an outstanding classification accuracy—reaching up to 100% in some tasks. This model outperformed other approaches and highlights the strong potential of ECG signals for enhancing model generalization.Extensive comparisons with the recent state of the art to verify the superiority of the proposed models.

The rest of this article is structured into four main sections. Section 2 provides a comprehensive overview of the research history related to state-of-the-art heart disease diagnosis using ECG images. Section 3 delves into the detailed description of the proposed framework for ECG image classification. Section 4 discusses the experimental results conducted, along with a comparison to recent related studies. Finally, Section 5 summarizes the key contributions of this work and highlights potential directions for future enhancements.

## 2. Literature Review

This section provides a comprehensive overview of recent advancements in heart disease diagnosis using ECG signal analysis, with DL and ML approaches. It highlights key studies that leverage CNNs, attention mechanisms, and hybrid DL models to improve classification accuracy. The review is structured to emphasize critical components such as preprocessing and augmentation techniques, segmentation strategies if applicable, and the integration of explainable AI and traditional ML classifiers. Table 1 summarized some of the recently published work using the ECG Images dataset of cardiac patients.

### 2.1. CNN-Based Methods

Recent developments in DL have revolutionized ECG image classification for cardiovascular disease diagnosis. Many researchers have proposed CNN-based architectures, often enhanced with attention mechanisms or hybrid designs, to classify ECG images with high accuracy. These studies focus on optimizing feature extraction from raw or preprocessed ECG signals to improve diagnostic precision. T. Sadad et al. [7] proposed a lightweight CNN integrated with attention modules and achieved a high classification accuracy of 98.39% on the ECG Images dataset of cardiac patients. Preprocessing steps such as brightness adjustment and resizing were incorporated. A similar approach using deep CNNs with normalization and data augmentation improved the performance to 97.47% [8]. An extended version using attention modules achieved an even higher accuracy of 98.73% [9]. However, these methods faced challenges with dataset imbalance and real-time deployment. Y. Liu et al. [10] presented a multimodal DL system that combined ECG images and textual records, achieving 99.63% accuracy using data augmentation. While effective, it may suffer from scalability issues when extending to real-time clinical systems. T. Alsayat [11] introduced an ensemble model of Inception, MobileNet, and NASNetLarge, reaching an F1 score of 0.9651 and balanced accuracy of 0.9640, though generalizability was limited by single-source data. R. Ao and G. He [12] implemented a VGG16-based model to diagnose 13 heart conditions from ECG images, reaching AUROC values up to 1.000 for specific conditions like RBBB, yet the binary classification design limited multi-label capabilities. A. H. Khan et al. [13] developed a cardiac disorder detection system using 12-lead ECG images from various formats. They proposed a generalized processing method and employs an SSD MobileNet v2-based deep neural network to detect four major cardiac abnormalities (myocardial infarction, abnormal heartbeat, prior MI history, and normal) with 98% accuracy. Leveraging the PhysioNet Challenge 2021 dataset, the model combined CNNs for spatial feature extraction and BiLSTM networks for temporal analysis, achieving peak performance metrics, including 99% accuracy for normal heartbeats and an overall accuracy of 98.5% [14].

### 2.2. Hybrid CNN and Machine Learning Models

Several research works combined DL with classical ML to enhance classification robustness and interpretability. These hybrid approaches often balance performance and computational efficiency while enabling better generalization. S. Reznichenko et al. [15] used CNNs and Random Forests on the PhysioNet 2020 dataset, achieving a macro F1 score of 0.761 for DL versus 0.759 for classical ML. While DL required fewer ECG leads, RF benefited from more inputs, indicating trade-offs between efficiency and complexity. The Fuzz-ClustNet model proposed in [16] incorporated fuzzy clustering with CNN-based feature extraction, achieving 98.66% and 95.79% accuracy on the MIT-BIH and PTB datasets, respectively. Despite strong results, the model’s clustering step increased the computational load. In [17], machine learning techniques trained on 2100 clinical records reached 90.1% training and 86.7% testing accuracies. While effective for early detection, the model’s reliance on clinical variables may limit its scalability to image-only diagnostics. The study in [18] introduced a lightweight CNN with a weighted ensemble of GNB, XGB, DT, and RF, achieving 98.93% accuracy. Although robust, the method required multiple classifiers, increasing implementation complexity. D. Alsekait et al. [19] developed Heart-Net, a multimodal DL model integrating MRI and ECG data using 3D U-Net and a temporal convolutional graph neural network (TCGNN). This approach achieved accuracies between 91.89% and 93.45% but introduced complexity due to its multimodal nature. S. Jayanthi and S. Devi [20] proposed AutoRhythmAI, which combined transformers with CNNs, achieving 97.39% and 96.60% for multi-class and binary classifications, respectively. However, it was computationally intensive. S. Sowmya and D. Jose [21] developed a CNN-LSTM hybrid model for arrhythmia detection, attaining 97% accuracy—slightly outperforming standalone CNNs. Another study used DenseNet201 for feature extraction, Relief and LASSO for selection, and deep residual shrinkage networks for classification, reaching 99.12% accuracy across five UCI datasets [22]. M. Muzammil et al. [23] examined the integration of AI, particularly DL-based CNNs, in ECG analysis for improved diagnosis and management of cardiovascular diseases. Various studies demonstrated AI’s high accuracy in detecting conditions such as atrial fibrillation and long QT syndrome, with area under the curve (AUC) values exceeding 0.85 and sensitivity and specificity rates surpassing 90%.

### 2.3. Transformer-Based and Attention-Enhanced Models

Recently, transformer architectures and attention mechanisms have emerged as powerful tools in image-based ECG signal analysis, offering enhanced capabilities in capturing long-range dependencies and emphasizing diagnostically relevant features within complex cardiac patterns. Y. Deng et al. [24] introduced DMSANet, a Deep Multi-Scale Attention Network, trained on PTB-XL to classify 15 cardiac conditions. The model achieved an average F1 score of 0.896 and showed excellent real-time monitoring capabilities. Another attention-based model was ECGTransForm [25], which employed BiTransformers with multiscale convolutions and achieved 99.26% accuracy on the MIT-BIH dataset using a hybrid of ResNet and LSTM. Though powerful, attention mechanisms can be resource-intensive and may require extensive tuning.

### 2.4. Explainable AI and Interpretable DL Techniques

To improve transparency and contextual awareness in ECG analysis, other studies explored explainable AI and multimodal learning, often incorporating clinical metadata or leveraging interpretability tools. For example, studies [26,27,28] developed interpretable DL systems using SHAP values and 1D CNNs on the CPSC2018 dataset, with average F1 scores reaching 0.813. These methods help reveal the importance of individual ECG leads, aiding clinical acceptance. However, they often rely on complex visualization methods that may not be suitable for time-sensitive diagnostics. In addition, the authors in [29] applied explainable ML models to a dataset of over 66,000 patients, where KNN achieved a 71% testing accuracy. SHAP was used for model transparency, identifying key risk factors. Although interpretable, the performance was lower than CNN-based models.

### 2.5. Transfer Learning and Custom Architectures

Several research works explored transfer learning using pretrained CNNs for image-based ECG classification, achieving a high performance even on limited data. However, dataset diversity, class imbalance, and real-world robustness remain concerns. M. Abubaker and B. Babayigit [30] explored SqueezeNet and AlexNet as feature extractors for traditional classifiers, with their custom CNN achieving 98.23% accuracy. H. Wu et al. [31] tackled digitization of paper-based ECGs for ML use and achieved over a 99% correlation with digital ECGs, though accuracy dropped with overlapping signals. O. Attallah [32,33] developed two deep ensemble models (ECG-BiCoNet and another CNN-based tool) for COVID-19 detection, achieving up to 98.8% binary and 91.73% multi-class accuracy, though performance was affected by a small dataset size and imbalance.

Several recent surveys [34,35,36] summarized ongoing advances in ECG-based disease diagnosis. These comprehensive reviews highlighted key trends, including the rise of hybrid DL models, data augmentation for imbalance handling, and increasing adoption of explainable AI techniques. They also emphasized the ongoing challenge of generalizability due to dataset limitations.

In our proposed approach, we employ pretrained CNN models to classify ECG signals into three categories: normal, history of myocardial infarction (MI), and abnormal. Our classification is performed on two types of data: the original image-based ECG signals and the reconstructed signals obtained after applying our segmentation method. This dual input enables us to evaluate the impact of segmentation on classification performance and enhances the robustness of the model across different signal representations.

**Table 1 diagnostics-15-01501-t001:** Recent research works based on ECG image datasets of heart disease patients.

Ref.	Year	Dataset	Dataset Size	Model	Accuracy
[13]	2021	ECG Images dataset	11,148 standard 12-lead-based ECG images	Single Shoot Detection (SSD) MobileNet v2-based DL	Multi-class diagnosis: 98 for 4 classes
[18]	2025	ECG Images dataset	1406 ECG images	CNN+ML	Transfer learning: 88.65% Transfer learning with GNB, SVM, and RF: 98.93%
[37]	2022	ECG Images dataset	928 ECG images (741 images for training and 187 images for testing)	MobileNetV2 and VGG16	MobileNetV2 model: 95% after fine-tuning VGG16 model: 95% after fine-tuning
[38]	2024	ECG Images dataset	928 ECG images	A novel image-vectorization method with ANNs	Binary classification models: Normal vs. MI: 92.39% Normal vs. Abn-HB: 88.88% Normal vs. His-MI: 78.72% Multi-label classification model: 89.58%
[39]	2022	ECG Images dataset	1682 ECG images	InceptionV3, ResNet50, MobileNetV2, VGG19, and DenseNet201	InRes-106 model: 98.34% InceptionV3 model: 90.56% ResNet50 model: 89.63% DenseNet201 model: 88.94% VGG19 model: 87.87% MobileNetV2 model: 80.56%

## 3. Methodology

The proposed three-phase framework for heart disease classification using ECG images is illustrated in Figure 1. The first phase, data cleaning, involves collecting raw ECG images and removing exact duplicate images. The dataset split into training and test sets was performed prior to ECG reconstruction to prevent any information leakage. The second phase, ECG signal segmentation, which focuses on isolating individual ECG leads to produce cleaner, more distinct signals, and the whole details are given in Section 3.1. In the final phase, feature extraction and classification, DL models like VGG16, VGG19, ResNet50, and InceptionNetV2 are used to extract features and classify reconstructed ECG images, with the whole details given in Section 3.2. This process leverages transfer learning and optional integration of SVM for an improved performance. The reconstructed ECG image, featuring distinct and separated signals, significantly improves signal clarity and boosts classification accuracy when using DL techniques.

### 3.1. ECG Signal Segmentation

Segmenting ECG signals is a challenging task due to the complex and highly variable nature of the signals. Overlapping waveforms and differences in signal shape—caused by individual health conditions or personal variations—make accurate segmentation difficult. On top of that, noise such as baseline drift and electrical interference can mask important features in the ECG. The need for real-time processing adds another layer of difficulty, as it limits algorithm complexity. Unpredictable patterns in pathological signals, like arrhythmias or other abnormalities, further complicate the process. To address these challenges, a proposed algorithm for ECG segmentation is presented and summarized in Algorithm 1. The algorithm follows a structured approach to extract ECG signals from images efficiently. Initially, preprocessing is performed where the ECG image is cropped to remove non-essential information, such as titles and patient details, ensuring that the focus remains solely on the signal data. To reduce noise and smooth the image, Gaussian blur is applied, which helps in improving the clarity of the signal without distorting the key features. Following preprocessing, the colored ECG image is binarized using Otsu’s method [40], enabling better differentiation of the ECG signal from the background noise, thus facilitating further analysis. To ensure methodological rigor, we confirm that all preprocessing, including denoising, segmentation, and enhancement, was performed solely on the training data.

The ECG lead separation step employs horizontal and vertical histograms to determine (1) lead baselines and (2) lead separations, thus segmenting the image into four distinct patches for individual lead segmentation. Typically, the four peaks of the horizontal histogram correspond to the lead baseline in the image, while the three vertical histogram peaks correspond to the three lead separation lines in the image.

To extract the lead signal from each patch, the algorithm uses the estimated baseline to identify a seed point—marking the start of the signal—by performing a vertical search around the baseline within a window of α pixels, as depicted in Figure 2a. Once the seed point is identified, the algorithm proceeds to scan the signal rightward, accounting for discontinuities and variations in pixel thickness. To improve efficiency, each subsequent signal point is located by searching within a β-pixel window around the vertical location of the previously identified point, thereby reducing search time, as shown in Figure 2b. Each detected signal point is recorded as its distance from the baseline. To address potential discontinuities in the isoelectric line—caused by various factors—if the algorithm fails to locate a signal point during the vertical search, it retains the value of the last successfully identified signal point immediately preceding the gap.
**Algorithm 1:** ECG signal segmentation**Input:** ECG_Image
**Output:** Segmented_ECG_Leads
**Parameters:**
   *σ*  ←Gaussian blur parameter
   α  ← Vertical search window size for seed point
   *β*  ← Vertical search window size for tracking signal
   *γ*  ← Margin for signal point validation
**1. Preprocessing:**
   **a.** Crop ECG_Image to remove borders and irrelevant regions.
    Cropped_image ← ECG_Image[280:1520, 150:2150]
   **b.** Smooth and reduce noise.
    Cropped_Image ← GaussianBlur(ECG_Image, σ)
**2. Image Binarization:**
   # Binary_Image is obtained by applying Otsu’s thresholding to ECG_Image.
   **a.**
Binary_Image ← Apply_Otsu_Thresholding(ECG_Image)
**3. ECG Lead Separation:**
   **a.** H_Histogram ← Compute_Horizontal_Histogram(Binary_Image)
   **b.** V_Histogram ← Compute_Vertical_Histogram(Binary_Image)
   **c.** Identify the top three vertical peaks in V_Histogram to define lead boundaries.
   V_Peak ← Sort(V_Histogram, reverse=true)[0:3]
   **d.** Divide Binary_Image into 4 image patches:
    # Patches correspond to Leads [1, 5, 9], Leads [2, 6, 10], Leads [3, 7, 11], and Leads [4, 8, 12]
   Patch_1←Binary_Image[0: Binary_Image.rows, 5:V_peaks[0]-5]
   Patch_2←Binary_Image[0:Binary_Image.rows,V_peaks[0]+5:V_peaks[1]-5]
   Patch_3←Binary_Image[0:Binary_Image.rows,V_peaks[1]+5:V_peaks[2]-5]
   Patch_4←Binary_Image[0:Binary_Image.rows,V_peaks[2]+5:Binary_Image.cols]
**4. Signal Extraction:**
   **For each** Patch in [Patch_1, Patch_2, Patch_3, Patch_4]:
   **a.** Identify the top four peaks in H_Histogram
    Baselines ← Sort(H_Histogram, reverse=true)[0:4]
   **b. For each **lead in Patch:
      i. Seed_Point ← Find_Seed_Point(Baselines[lead], window = α)
      ii. Signal ← Initialize signal as an empty list

      iii. Current_Point ← Seed_Point
      iv. While Current_Point < Patch width:
         - Next_Window ← Vertical_Search_Window(Current_Point, size = β)
         - Next_Point ← Track_Signal_Point(Next_Window)
         # Check for discontinuity 
         - If Discontinuity_Detected(Next_Point):
          Next_Point←Signal[Signal.length - 1]
         # Check for multiple point detection
         - If Thick_Isoelectric_Line_Detected:
          Next_Point← Estimate_Median_Point(identified points)
         # Validate the point w.r.t the previously added point 
         - If Validate_Point(Next_Point, Baseline, *γ*):
            Signal.append(Next_Point)
            Current_Point ← Next_Point
     **c.** Store Signal in Segmented_ECG_Leads**Return** Segmented_ECG_Leads

During the vertical search for a signal point, multiple pixels may be detected due to the thickness of the isoelectric line; the pixels are highlighted in orange in Figure 2b. To handle this, the algorithm computes the median of the identified pixels, as it is more robust to outliers than the average. In the last step, each candidate point is validated before being accepted as part of the signal to further reduce outliers. Specifically, the algorithm checks that the distance of the candidate point from the baseline falls within a margin of *γ* pixels relative to the previously identified point, Figure 2c. As a result, the length of the extracted signal corresponds to the full width of the patch.

### 3.2. Pretrained DL Models

This section introduces a brief description of the pretrained DL models addressed in this paper.

*InceptionV2* is an improved version of the original Inception (GoogLeNet) DL model, designed to enhance computational efficiency and accuracy in image classification tasks. It introduces key optimizations, such as batch normalization, factorized convolutions, and reduced parameter count, which help mitigate the vanishing gradient problem and improve training stability. By decomposing large convolutional filters into smaller ones (e.g., a 3 × 3 filter split into two 3 × 1 and 1 × 3 filters), InceptionV2 reduces the number of parameters and computational cost while maintaining high representational power. These refinements make it a strong choice for large-scale image recognition applications, including medical imaging, object detection, and autonomous systems [41].

*ResNet50* is a DL model that belongs to the ResNet (Residual Network) family, introduced by He et al. in 2016 [42]. It is a 50-layer deep convolutional neural network designed to address the vanishing gradient problem in very deep networks by utilizing residual connections, or “skip connections” These connections help preserve gradient flow and enable efficient training of deep architectures. ResNet50 consists of convolutional layers, batch normalization, and ReLU activations, along with bottleneck residual blocks that improve computational efficiency. Due to its strong feature extraction capabilities, ResNet50 is widely used in computer vision tasks such as image classification, object detection, and segmentation. It has been pretrained on large datasets like ImageNet, making it a popular choice for transfer learning applications.

*VGG16 and VGG19* are deep convolutional neural network architectures introduced by the Visual Geometry Group (VGG) at the University of Oxford. Both models were developed for image classification tasks and gained prominence through their performance in the ImageNet Large Scale Visual Recognition Challenge (ILSVRC) 2014. VGG16 consists of 16 layers, while VGG19 has 19 layers, both following a consistent pattern of small 3 × 3 convolutional filters stacked sequentially, followed by max-pooling layers and fully connected layers at the end. These architectures emphasize depth with a uniform design, making them computationally expensive but effective for feature extraction and transfer learning. Despite their higher parameter count compared to more modern architectures, VGG networks remain widely used for pretrained models in DL applications [43].

*GoogleNet*, introduced by Szegedy et al. in 2016 [44], is a deep convolutional neural network architecture that won the ImageNet Large Scale Visual Recognition Challenge (ILSVRC) that year. It is based on the Inception module, which improves computational efficiency by using multiple convolutional filter sizes in parallel and reducing the number of parameters through 1 × 1 convolutions. This design allows for deeper networks without a significant increase in computational cost. GoogleNet consists of 22 layers and employs global average pooling instead of fully connected layers to reduce overfitting. Its inception architecture has inspired many subsequent DL models.

Pretrained models are typically trained on large, diverse datasets (e.g., ImageNet), allowing them to learn high-level features such as edges, shapes, and textures. These models are then adapted (through transfer learning) to new tasks like ECG classification. By leveraging the knowledge from these pretrained models, we can avoid the need to train a model from scratch and speed up the model development process. Pretrained models like VGG16, VGG19, ResNet50, GoogleNet, and Inceptionv2 are designed to extract robust and hierarchical features from input data. Even though these models were originally designed for image data, their deep layers can extract meaningful features from the time-domain representations of ECG signals. These models are built with deep architectures consisting of many layers. The deep layers in these models help in capturing intricate patterns in ECG signals, which could be missed by shallow architectures. By using those models, the development time is reduced, and computational resources are saved, allowing for faster prototyping and deployment of the ECG classification system. Moreover, they reduce the likelihood of overfitting on small ECG datasets, especially when fine-tuned using a small number of ECG samples. Using these proven models can lead to higher accuracy and reliability in ECG signal classification, improving diagnostic capabilities. The flexibility of using those models on 2D representations of ECG signals enables their direct application to ECG classification tasks without requiring significant changes to the model architecture. These models help extract valuable features from ECG signals, which can lead to more reliable and effective heart disease diagnostics and anomaly detection.

The pretrained models used in our experiments had nearly identical parameter settings, with only minor differences in the maximum number of epochs, as shown in Table 2. Across all models, the mini-batch size was either 10, 20, or 50, and the initial learning rate was set at 1 × 10^−^⁴. The solver used in all cases was stochastic gradient descent with momentum (SGDM), ensuring consistent optimization. The primary differentiation among the models was the maximum number of epochs, which varied based on the complexity of the architecture. For example, InceptionV2, ResNet50, and VGG16 were trained for 40, 20, and 60 epochs, respectively. This suggests that deeper models required longer training durations to fully capture ECG features, whereas shallower models reached convergence faster. These parameter choices played a role in determining the effectiveness of each model in classification tasks.

## 4. Results and Discussion

This section provides a thorough evaluation of the proposed framework for classifying heart disease using ECG images. The experiments were carried out on a carefully curated dataset of 12-lead ECG images, as described in Section 4.1. Section 4.2 details each step of the ECG segmentation process along with its performance evaluation. Section 4.3 presents the classification results from various DL models, covering both binary and multi-class tasks. The analysis also explores how the ECG segmentation process affects model performance, with comparisons across different architectures to assess their ability to classify the reconstructed ECG signal image. Finally, the framework is benchmarked against recent state-of-the-art methods, demonstrating its superior accuracy and robustness.

The experiments were implemented on an Alienware Aurora R9 desktop, equipped with an octa-core Intel^®^ Core™ i9-9900 processor with 3.10 GHz, 32 GB RAM, and a 1 TB hard disk drive. The implementation code runs under the Microsoft Windows 10 Home (×64) operating system and is written in MATLAB 2021b with some related toolboxes.

### 4.1. Dataset Description

The dataset utilized in this paper comprises 12 lead-based standard ECG images collected from distinct patients from diverse cardiac institutes across Pakistan. The ECG images do not contain any personal information about the patient. All ECG images have been annotated by several medical experts [45].

The dataset, as described [45], comprises four primary classes: normal, myocardial infarction (MI), abnormal, and history of MI. In the data-cleaning phase, we identified a high frequency of duplicated images within the myocardial infarction class, which compromised its integrity. As a result, we excluded this class from our experiments. For the remaining classes (normal, history of MI, and abnormal), we performed a cleaning process that involved removing exact duplicate images and discarding samples with interlaced or overlapping ECG signals that could not be reliably separated for analysis. Table 3 presented the number of images for the different categories.

A normal person, in medical terms, functions naturally without observable abnormalities or deficiencies. MI, commonly known as a heart attack, occurs when blood flow to a part of the heart is reduced or blocked, causing severe damage, often presenting as chest pain or discomfort radiating to the shoulder, arm, back, neck, or jaw. Patients with a previous history of MI are those that have recently recovered from a heart attack. Figure 3 displays a sample for each category.

### 4.2. Evaluation Metrics

In this work, standard evaluation metrics are used to assess the classification performance of the proposed framework. These metrics include validation *accuracy*, *precision*, *recall*, and *F1 score*, which are calculated as follows:*Accuracy*: This metric is defined based on the values of true positives (*TPs*), true negatives (*TNs*), false negatives (*FNs*), and false positives (*FPs*) and is represented by Equation (1).(1)$$Accuracy=\frac{TP+TN}{TP+TN+FP+FN}$$

*Precision*: This measure is the ratio of true positives to true and false positives and is shown in Equation (2).


(2)
$$Precision=\frac{TP}{TP+FP}$$


*Recall*: This metric is measured as the ratio of correctly identified positive examples to the total number of positive models. Equation (3) shows the recall evaluation measure.


(3)
$$Recall=\frac{TP}{TP+FN}$$


*The F1 score***:** This measure is a useful metric in ML and DL, which is a combination of two other metrics. Equation (4) shows the F1 score measure evaluation.

(4)$$F1\;Score=2\times \frac{Precision\times Recall}{Precision+Recall}$$
where TP, TN,FP, and FN are the true positive, true negative, false positive, and false negative, respectively.

### 4.3. ECG Signal Segmentation Evaluation

Table 4 shows the applied tuning parameter of the proposed ECG signal segmentation algorithm, including values for the vertical search windows (*α* and *β*), the validation margin (*γ*), and other key thresholds. These parameters were selected based on empirical testing to balance accuracy and computational efficiency. Each parameter plays a critical role in guiding the signal extraction process, such as ensuring robustness against noise, accommodating line thickness variations, and reducing false detections.

The first step in the segmentation algorithm is preprocessing, where the ECG image is cropped and smoothed with Gaussian blur. Figure 4 displays an example of a preprocessed signal and its corresponding binarized image.

The next step is the signal separation based on histograms. A horizontal histogram is constructed to identify the baseline of the ECG signal, with peaks corresponding to isoelectric lines. Similarly, a vertical histogram is used to determine lead separations, with the top three peaks indicating the separation lines. Based on these peaks, the image is divided into three vertical patches, each containing specific leads (e.g., Patch 1: Leads 1, 5, 9). To avoid interference from text annotations indicating lead names, these segments are cropped out before proceeding. Figure 5 depicts the horizontal and vertical histograms of the extracted signal. Figure 6 shows the signal leads’ separation of the extracted signal and the corresponding patches. The top row of the figure shows the estimated baseline corresponding to isoelectric lines in blue color and the vertical separation line in red color.

The signal extraction step begins with obtaining four baseline estimates from the horizontal histogram for a single patch, where each baseline corresponds to a single lead. The process starts by identifying a seed point near the baseline using a vertical scan. Afterwards, the signal is scanned iteratively in the right direction, using a search window around the amplitude of the previous point to locate the next signal point. This process results in a signal whose length matches the width of the patch. Figure 7 shows the extracted lead signals from patch 1 of the ECG image presented in Figure 6.

The segmented ECG signals were evaluated against the original dataset images, which served as the ground truth. A synthesized image was generated using the extracted signals, and its quality was assessed by calculating the Structural Similarity Index (SSIM) and Mean Squared Error (MSE) compared to the original ECG image, as defined in Equations (5) and (26) [46].(5)$$SSIM\left(x,y\right)=\frac{\left(2\mu_{x}\mu_{y}+C_{1})(2\sigma_{xy}+C_{2}\right)}{\left(\mu_{x}^{2}+\mu_{y}^{2}+C_{1})(\sigma_{x}^{2}+\sigma_{y}^{2}+C_{2}\right)}$$
where *x* and *y* represent two signals being compared; *μ_x_* and *μ_y_* are the mean of *x* and *y*; *σ_x_*^2^ and *σ_y_*^2^ refer to the variances of *x* and *y*; *σ_xy_* refers to the covariance of *x* and *y*; and $C_{1}$ and $C_{2}$ are small constants to avoid division by zero, typically defined as $C_{1}={\left(K_{1}L\right)}^{2},\;C_{2}={\left(K_{2}L\right)}^{2}$.(6)$$MSE=\frac{1}{N}\sum_{i=1}^{N}{\left(y_{i}-{\hat{y}}_{i}\right)}^{2}$$
where $y_{i}$ and ${\hat{y}}_{i}$ refer to the true value and the predicted value at point *i*, respectively, and *N* is the total number of data points.

Figure 8 illustrates the alignment between the binarized ECG image (shown in red) and the synthesized image of the segmented leads (shown in cyan) for patches 1, 2, and 3 of the ECG images presented in Figure 6. SSIM and MSE values were calculated across all images from the three dataset classes—normal, history of MI, and abnormal—resulting in overall alignment scores of 0.94 and 0.002 for SSIM and MSE, respectively. It is worth noting that variations in signal thickness between the original and synthesized lead images affected the SSIM value. Additionally, Figure 9 demonstrates several challenging ECG signal examples alongside their corresponding reconstructed signals.

### 4.4. ECG Signal Classification

This section demonstrates the effectiveness of the proposed ECG image segmentation and DL-based classification framework. The performance of various DL models was evaluated in terms of accuracy, precision, recall, and F1 score across different classification tasks, including binary and multi-class classification.

The ECG segmentation process significantly improved signal clarity and feature extraction. The results demonstrated that the segmentation algorithm successfully separated ECG leads and extracted relevant signals, thereby improving the performance of classification models. This improvement in segmentation contributed to enhanced accuracy in classification tasks, making it easier to distinguish between normal and abnormal ECG signals.

#### 4.4.1. Binary Classification Results

Table 5, Table 6 and Table 7 provide insights into the performance of different models when distinguishing between ECG categories. In the abnormal vs. myocardial infarction (MI) classification, models demonstrated high accuracy, with VGG19 + SVM achieving the best performance using the reconstructed ECG dataset, reaching an accuracy of 98.51%, as illustrated in Figure 10 and Table 5, which demonstrate the accuracy of all models. This indicates the effectiveness of segmentation in distinguishing abnormal heartbeat patterns from MI cases. Table 6 presents the binary classification of normal vs. MI classification using the different pretrained models, including VGG16, VGG16 + SVM, and InceptionNetV2, which achieved a perfect classification accuracy of 100%, demonstrating that distinguishing between normal ECGs and MI patients is a relatively straightforward task.

In the normal vs. abnormal classification, as presented in Table 7, VGG19 exhibited a high accuracy, scoring 99.34% using the original ECG dataset and 98.03% using the reconstructed ECG dataset, highlighting the reliability of DL models in identifying abnormal heartbeat patterns.

#### 4.4.2. Multi-Class Classification Results

Table 8 further demonstrates the competitive performance of the proposed models. According to the results declared in the table, VGG19 + SVM achieved 100% accuracy, outperforming the other models, emphasizing the capability of the proposed framework in distinguishing between multiple ECG categories effectively. This result is particularly significant, as multi-class classification poses more complexity than binary classification due to the overlapping nature of some ECG conditions. Figure 11 depicts the accuracies of the pretrained models addressed in this study.

Figure 12, Figure 13, Figure 14, Figure 15 and Figure 16 present confusion matrices for different transfer learning models used in ECG classification, providing a visual representation of each model’s performance in distinguishing between ECG categories. Each confusion matrix highlights its classification accuracy on both the original and the reconstructed datasets. Collectively, these matrices offer valuable insights into the models’ true positive, true negative, false positive, and false negative rates, essential for evaluating their effectiveness in ECG signal classification.

#### 4.4.3. Comparative Performance Evaluation with Recent Related Work

A comparison of the proposed models with some related studies, such as the work in [13,18,37,38,39], highlights the superior performance of the proposed framework, as shown in Table 9. The results indicate that the proposed models achieved a higher accuracy in both binary and multi-class classifications compared to existing methodologies. Notably, the VGG19+SVM model achieved an accuracy of 100% and surpassed models such as SSDMobileNetV2 [13], MobileNetV2 [37], ANNs [38], and InceptionV3 [39] in multi-class diagnosis, demonstrating the advantages of the enhanced segmentation technique. Among the related published work, the best achievement is proposed in [18], achieving an accuracy of 98.93% for multi-class classification. The superior performance of these models reinforces the importance of effective preprocessing techniques and robust DL architectures in ECG classification. Figure 17 depicts the best accuracy obtained by the proposed model against some recent related work.

**Table 9 diagnostics-15-01501-t009:** Performance comparisons of the proposed models against related works.

Reference	Model Used	Average Accuracy			
		Binary Classification			Multi-Class Classification
		Normal vs. Abnormal	Normal vs. MI	Abnormal vs. MI	
[18]	InceptionV3 + GNB	-	-	-	92.91%
	ResNet50 + RF	-	-	-	92.19%
	DenseNet169 + RF	-	-	-	92.55%
	lightweight CNN	-	-	-	98.93%
[13]	SSDMobileNetV2	97.25%	100%	96.3%	98.33%
[37]	MobileNetV2	-	-	-	95%
	VGG16	-	-	-	95%
[38]	ANNs	92.39%	98.26%	-	89.58%
[39]	InceptionV3	-	-	-	90.56%
	ResNet50	-	-	-	89.63%
	MobileNetV2	-	-	-	80.56%
	VGG19	-	-	-	87.87%
	DenseNet201	-	-	-	88.94%
Proposed (using reconstructed ECG dataset)	VGG16	98.08%	100%	97.46%	96.58%
	VGG16 + SVM	98.08%	100%	97.76%	97.76%
	InceptionNetv2	93.14%	100%	88.46%	95.56%
	InceptionNetv2 + SVM	96.67%	98.25%	93.33%	95.76%
	VGG19	98.03%	100%	98.29%	99.51%
	VGG19 + SVM	97.33%	100%	98.51%	100%
	ResNet50	96.05%	88.99%	92.31%	90.10%
	ResNet50 + SVM	95.52%	91.29%	91.76%	90.77%
	GoogleNet	95.39%	96.30%	94.87%	90.10%
	GoogleNet + SVM	95.41%	94.82%	97.01%	90.40%

#### 4.4.4. Discussion About the Achieved Results

A comparison between the results obtained using the original dataset and the reconstructed dataset reveals that while most models performed well on both, there were variations in accuracy across different classification tasks. Notably, VGG19+ SVM and VGG16 + SVM consistently performed well on both datasets, maintaining a high classification accuracy. In contrast, models such as InceptionNetV2 showed a more significant drop in accuracy when applied to the reconstructed dataset, indicating a potential reliance on features that were more prominent in the original dataset but less distinct after segmentation. This suggests that deeper models with strong feature extraction capabilities, particularly VGG-based architectures, are better suited for handling reconstructed ECG images, while models like InceptionNetV2 may require further optimization to adapt to refined datasets.

The results also highlight the superior performance of certain transfer learning models in ECG classification, with VGG19 + SVM emerging as the most effective model. It consistently outperformed other architectures across multiple classification tasks, achieving a high accuracy in both binary and multi-class classifications. Specifically, VGG19 + SVM demonstrated a 98.51% accuracy in abnormal vs. myocardial infarction (MI) classification, 100% accuracy in normal vs. MI classification, and 99.34% accuracy in normal vs. abnormal classification. These results confirm its ability to accurately differentiate between ECG categories.

The superiority of VGG19 + SVM stems from its deep feature extraction capabilities combined with SVM’s enhanced classification strategy. VGG19′s multiple convolutional layers effectively capture complex ECG patterns, while SVM optimizes decision boundaries, leading to improved generalization and robustness. Furthermore, the enhanced ECG segmentation technique significantly contributed to the model’s performance by refining input data, reducing noise interference, and ensuring high-quality feature extraction. On the other hand, some models exhibited weaknesses that limited their classification accuracy. InceptionNetV2 showed lower accuracy in abnormal vs. MI classification (88.46%%), likely due to its sensitivity to waveform variations, which affects its ability to capture subtle ECG differences. Overall, the findings demonstrate that deeper and more sophisticated models, particularly VGG19 + SVM, are better suited for ECG classification due to their superior feature extraction and classification capabilities. The dataset size (674 images) is considered a limitation. It may have a potential impact on generalizability, and we have future plans to validate the model on larger and more diverse datasets, including real-time clinical ECG recordings.

## 5. Conclusions

This study has introduced a comprehensive and robust framework for heart disease diagnosis by leveraging ECG signal segmentation and deep transfer learning classification, augmented with optional SVM integration. The proposed methodology effectively addresses key challenges in ECG image analysis, including waveform variability, noise interference, and overlapping signals. By employing a novel ECG signal segmentation algorithm, we significantly enhanced signal clarity, enabling more precise feature extraction and classification. The framework demonstrated an exceptional performance across both binary and multi-class classification tasks, with the VGG19 + SVM model achieving perfect accuracy in several evaluations (an accuracy of 98.01% and 100% accuracy in multi-class classification and average accuracies of 99% and 97.95 in binary tasks based on original and reconstructed datasets, respectively). This highlights the critical impact of preprocessing and segmentation on the effectiveness of transfer learning in medical image analysis. Furthermore, extensive comparative analysis against recent state-of-the-art techniques affirmed the superiority of our framework in terms of accuracy, robustness, and generalization capabilities. The contributions of this work highlight the significant impact of AI in advancing clinical diagnostics and set the foundation for more accurate and automated ECG-driven cardiovascular disease detection systems.

This study addresses future applications of the proposed ECG signal reconstruction and classification framework, particularly in the context of real-time mobile ECG analysis and clinical deployment. It suggests that the DL models, especially when enhanced by effective ECG segmentation and denoising, offer the potential for integration into wearable health monitoring systems and point-of-care diagnostic tools. These advances could support real-time, automated heart disease diagnosis using mobile devices, improve remote patient monitoring, and reduce the need for expert manual ECG interpretation. Additionally, the framework holds promise for clinical deployment as a decision support tool for cardiologists. By assisting in the interpretation of ECG signals and highlighting abnormal patterns, it can reduce the diagnostic time and support more accurate assessments, especially in resource-limited settings. These applications collectively highlight the model’s potential impact in advancing both personal and clinical cardiac care.

## Figures and Tables

**Figure 1 diagnostics-15-01501-f001:**
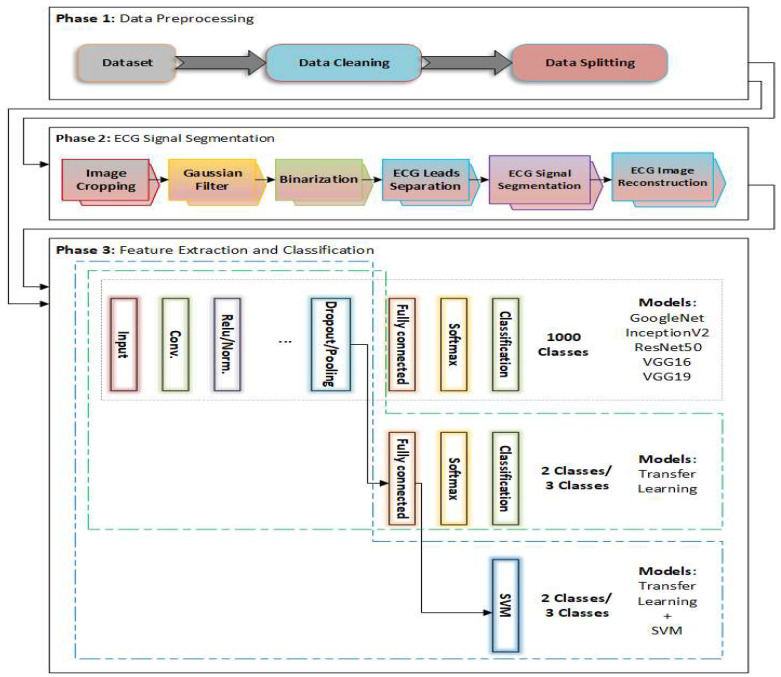
The proposed framework for ECG image classification.

**Figure 2 diagnostics-15-01501-f002:**
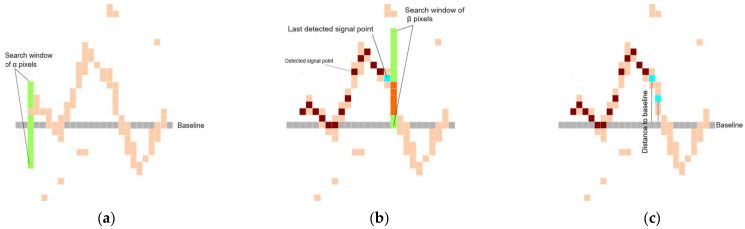
ECG signal extraction algorithm. (**a**) Estimated horizontal baseline (grey) over the ECG signal with the seed point search window shown in green. (**b**) Detection of signal pixels progressing in the rightward direction. (**c**) Validation of each newly detected signal point based on its position relative to the previously detected point.

**Figure 3 diagnostics-15-01501-f003:**
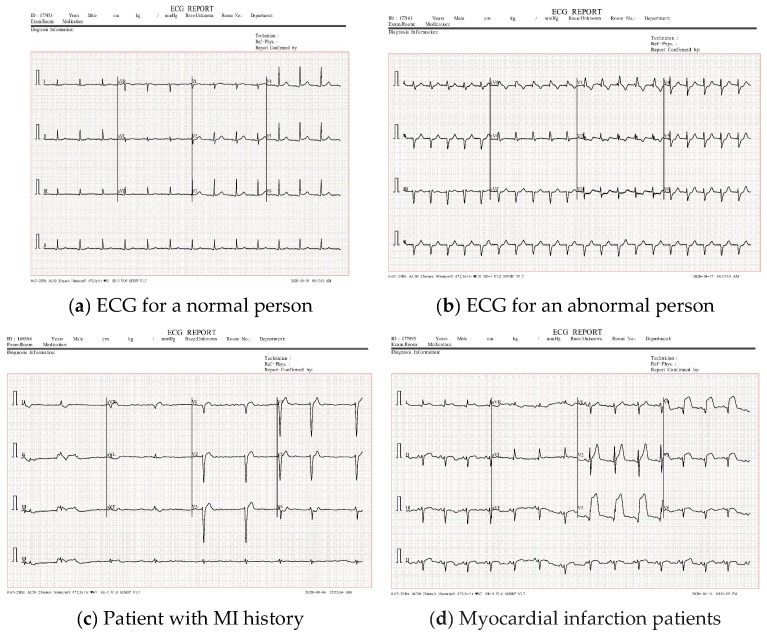
Samples of ECG dataset categories.

**Figure 4 diagnostics-15-01501-f004:**
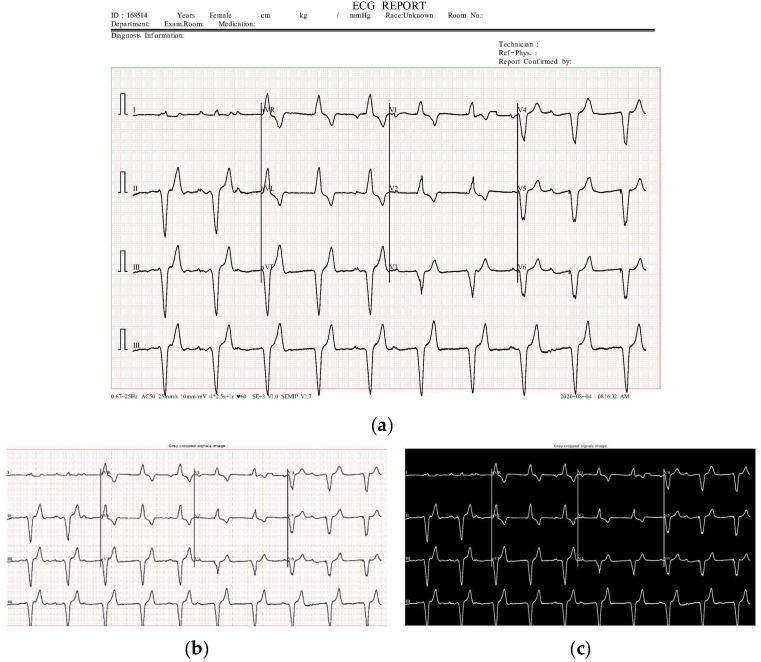
A sample of a cropped image and its corresponding binarization. The top row (**a**) displays the original image from the dataset, while the bottom row shows the cropped image on the left (**b**) and its binarized counterpart on the right (**c**).

**Figure 5 diagnostics-15-01501-f005:**
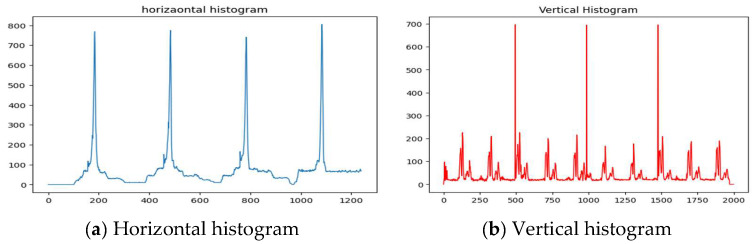
The horizontal and vertical histograms of the extracted signal.

**Figure 6 diagnostics-15-01501-f006:**
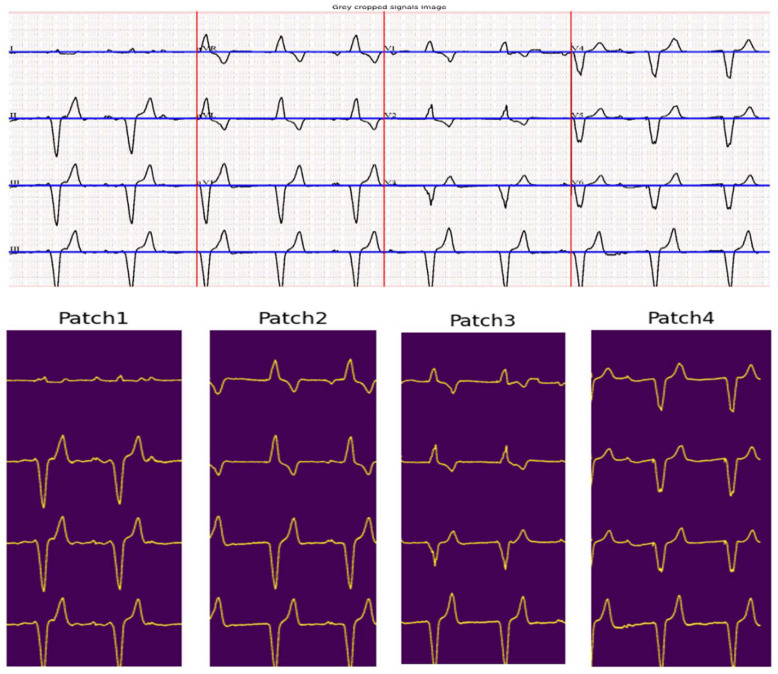
Separation of ECG leads and their corresponding segmented patches. The top row shows the projections of the detected histogram peaks—horizontal peaks in blue and vertical peaks in red—overlaid on the original ECG image. The bottom row displays the resulting segmented image patches.

**Figure 7 diagnostics-15-01501-f007:**
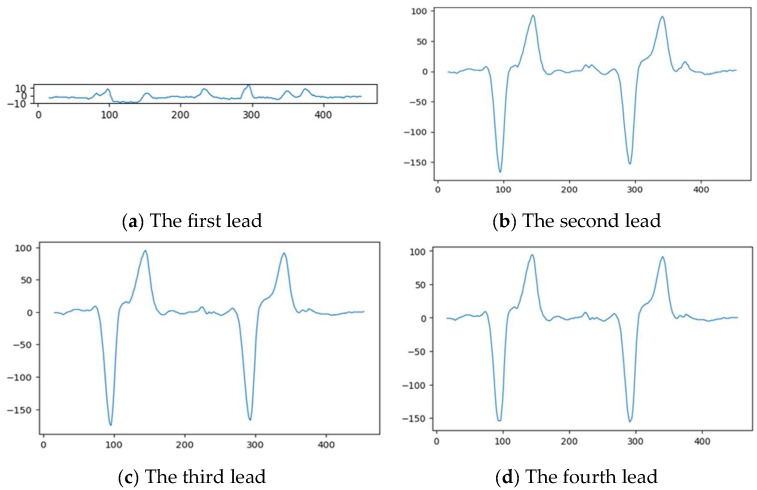
The extracted leads from patch 1.

**Figure 8 diagnostics-15-01501-f008:**
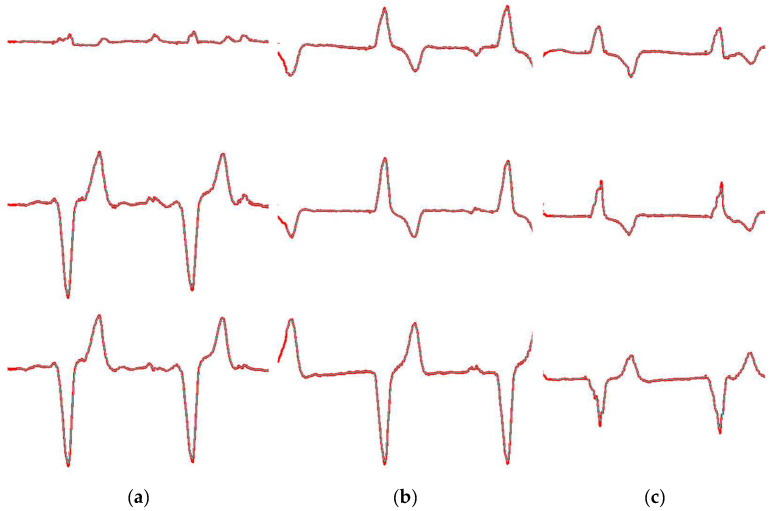
Alignment between the binarized ECG image and the segmented leads obtained from Algorithm 1: (**a**) Alignment of patch 1: The ECG binarized image is shown in red, and the segmented signal in cyan. (**b**) Alignment of patch 2: The ECG binarized image is shown in red, and the segmented signal in cyan. (**c**) Alignment of patch 3: The ECG binarized image is shown in red, and the segmented signal in cyan.

**Figure 9 diagnostics-15-01501-f009:**
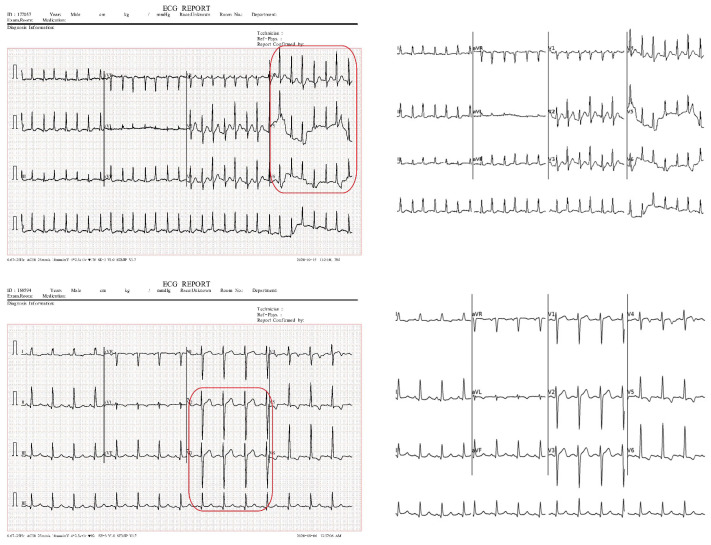
Examples of challenging ECG signals and their corresponding reconstructed signals. The red rectangles highlight the challenging segments successfully processed by the segmentation algorithm, including those with random signal artifacts or signals positioned very closely.

**Figure 10 diagnostics-15-01501-f010:**
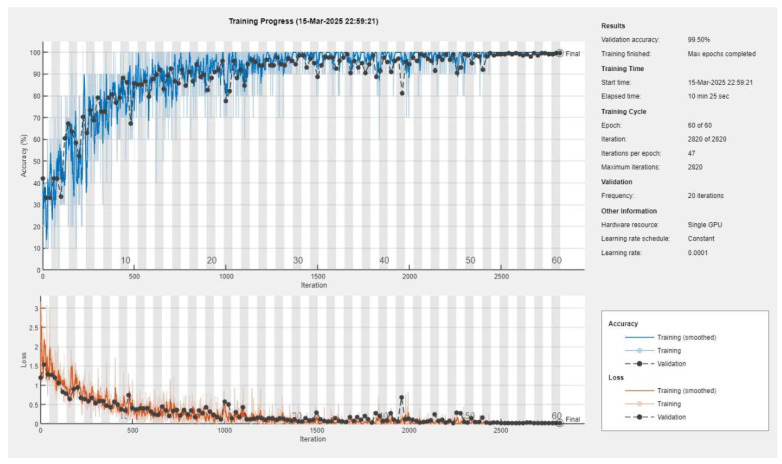
Training and loss accuracies of VGG19 + SVM using the reconstructed dataset.

**Figure 11 diagnostics-15-01501-f011:**
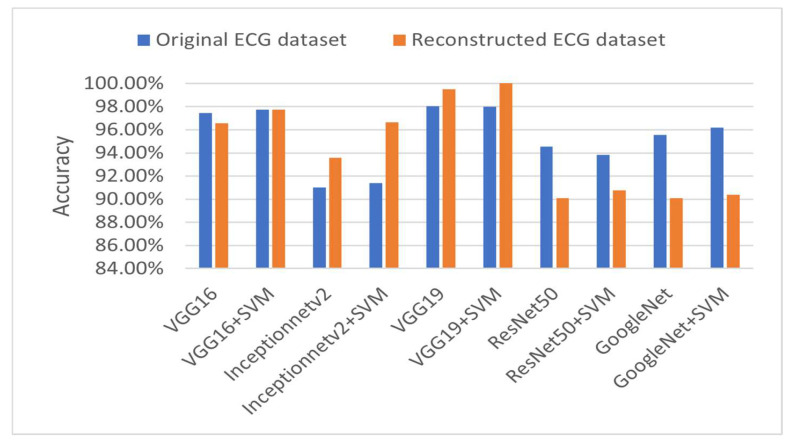
Multi-class accuracies of the proposed pretrained models.

**Figure 12 diagnostics-15-01501-f012:**
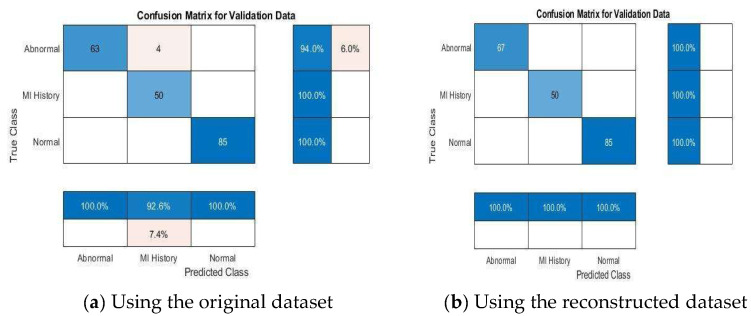
Confusion matrix of VGG19 model using original and reconstructed datasets.

**Figure 13 diagnostics-15-01501-f013:**
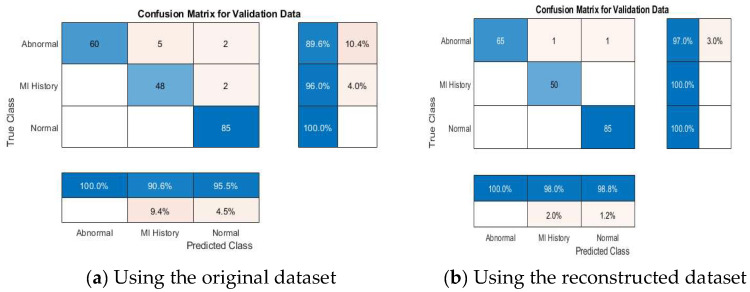
Confusion matrix of VGG16 model using original and reconstructed datasets.

**Figure 14 diagnostics-15-01501-f014:**
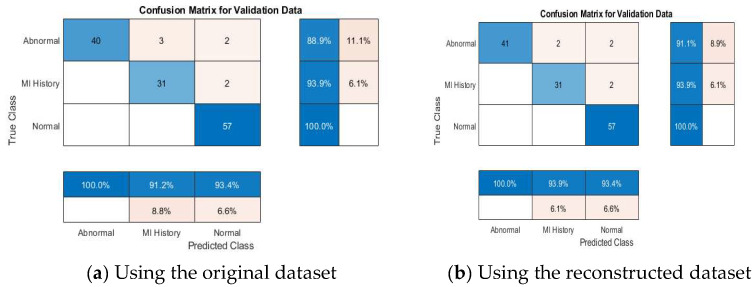
Confusion matrix of the InceptionNetv2 model using original and reconstructed datasets.

**Figure 15 diagnostics-15-01501-f015:**
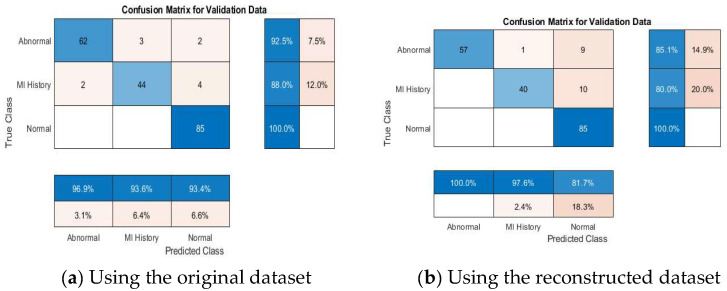
Confusion matrix of the ResNet50 model using original and reconstructed datasets.

**Figure 16 diagnostics-15-01501-f016:**
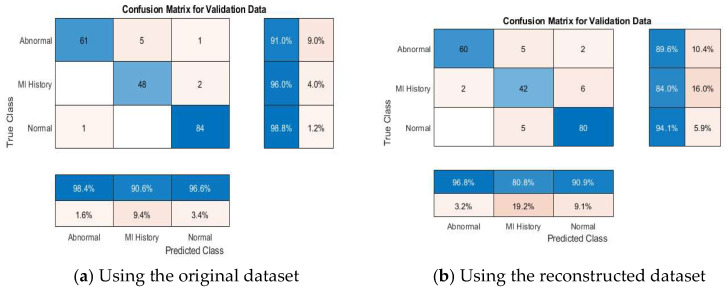
Confusion matrix of the GoogleNet model using original and reconstructed datasets.

**Figure 17 diagnostics-15-01501-f017:**
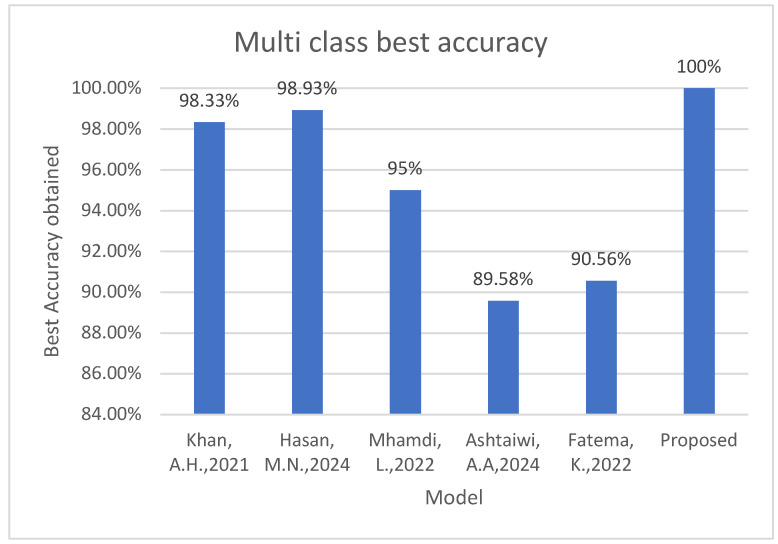
The best accuracy obtained by the proposed model against some recently related work [13,18,37,38,39].

**Table 2 diagnostics-15-01501-t002:** The parameters of the pretrained models.

	Training Solver	Mini Batch Size	Max Epochs	Initial Learn Rate	Validation Frequency
InceptionNetV2	SGDM	10	40	1 × 10^−4^	20
ResNet50	SGDM	20	20	1 × 10^−4^	20
VGG16	SGDM	10	60	1 × 10^−4^	20
VGG19	SGDM	10	60	1 × 10^−4^	20
GoogleNet	SGDM	50	100	1 × 10^−4^	60

**Table 3 diagnostics-15-01501-t003:** ECG image dataset categories and number of images in each category.

No.	Category	Training Set	Testing Set	Total ECG Images
1	Normal	199	85	284
2	History of MI	116	50	166
3	Abnormal	157	67	224
4	Total dataset	472	202	674

**Table 4 diagnostics-15-01501-t004:** Parameters tuning of Algorithm 1.

Parameter	Value
Gaussian blur (σ)	0.7
Search window for seed point (*α*)	150 pixels
Search window for subsequent signal point (*β*)	75 pixels
Validating candidate signal point (*γ*)	15 pixels

**Table 5 diagnostics-15-01501-t005:** Binary classification (abnormal and MI) results using the original and reconstructed ECG datasets.

	Results Using the Original ECG Dataset				Results Using the Reconstructed ECG Dataset			
	Overall Precision	Overall Recall	F1 Score	Accuracy	Overall Precision	Overall Recall	F1 Score	Accuracy
VGG16	97.17%	97.76%	97.46%	97.44%	97.17%	97.76%	97.46%	97.44%
VGG16 + SVM	97.17%	97.76%	97.46%	97.76%	97.17%	97.76%	97.46%	97.76%
InceptionNetv2	94.59%	95.56%	95.07%	94.87%	88.68%	89.60%	89.14%	88.46%
InceptionNetv2 + SVM	94.59%	95.56%	95.07%	95.56%	92.31%	93.33%	92.82%	93.33%
VGG19	97.46%	97.76%	97.17%	97.44%	98.08%	98.51%	98.29%	98.29%
VGG19 + SVM	97.17%	97.76%	97.46%	97.76%	98.08%	98.51%	98.29%	98.51%
ResNet50	94.00%	93.76%	93.88%	94.02%	92.52%	91.76%	92.14%	92.31%
ResNet50 + SVM	96.51%	96.51%	96.51%	96.51%	91.49%	91.01%	91.25%	91.76%
GoogleNet	95.45%	96.27%	95.86%	95.73%	94.64%	95.52%	95.08%	94.87%
GoogleNet + SVM	96.30%	97.01%	96.65%	97.01%	96.30%	97.01%	96.65%	97.01%

**Table 6 diagnostics-15-01501-t006:** Binary classification (normal and MI) results using the original and reconstructed ECG datasets.

	Results Using the Original ECG Dataset				Results Using the Reconstructed ECG Dataset			
	Overall Precision	Overall Recall	F1 Score	Accuracy	Overall Precision	Overall Recall	F1 Score	Accuracy
VGG16	100%	100%	100%	100%	100%	100%	100%	100%
VGG16 + SVM	100%	100%	100%	100%	100%	100%	100%	100%
InceptionNetv2	100%	100%	100%	100%	100%	100%	100%	100%
InceptionNetv2 + SVM	100%	100%	100%	100%	97.14%	98.25%	97.69%	98.25%
VGG19	100%	100%	100%	100%	98.85%	98%	98.42%	98.52%
VGG19 + SVM	100%	100%	100%	100%	98.85%	98%	98.42%	98%
ResNet50	98.85%	98%	98.42%	98.52%	89.63%	88.69%	89.29%	88.99%
ResNet50 + SVM	98.85%	98%	98.42%	98%	90.12%	91.29%	90.71%	91.29%
GoogleNet	100%	100%	100%	100%	95.45%	97.06%	96.25%	96.30%
GoogleNet + SVM	100%	100%	100%	100%	95.62%	94.82%	95.22%	94.82%

**Table 7 diagnostics-15-01501-t007:** Binary classification (normal and abnormal) results using the original and synthetic ECG datasets.

	Using the Original ECG Dataset				Using the Reconstructed ECG Dataset			
	Overall Precision	Overall Recall	F1 Score	Accuracy	Overall Precision	Overall Recall	F1 Score	Accuracy
VGG16	97.93%	98.08%	98.01%	98.03%	97.93%	98.08%	98.01%	98.03
VGG16 + SVM	97.93%	98.08%	98.01%	98.08%	97.93%	98.08%	98.01%	98.08%
InceptionNetv2	96.29%	95.79%	96.04%	96.08%	94.53%	92.22%	93.36%	93.14%
InceptionNetv2 + SVM	97.87%	98.25%	98.06%	95.79%	97.50%	96.67%	97.08%	96.67%
VGG19	99.42%	99.25%	99.34%	99.34%	97.93%	98.08%	98.01%	98.03%
VGG19 + SVM	99.42%	99.25%	99.34%	99.25%	97.33%	97.33%	97.33%	97.33%
ResNet50	98.85%	98.51%	98.68%	98.68%	96.70%	95.52%	96.11%	96.05%
ResNet50 + SVM	98.30%	97.76%	98.03%	97.76%	96.70%	95.52%	96.11%	95.52%
GoogleNet	96.16%	95.84%	96.00%	96.05%	95.27%	95.41%	95.34%	95.39%
GoogleNet + SVM	96.74%	96.58%	96.66%	96.58%	95.27%	95.41%	95.34%	95.41%

**Table 8 diagnostics-15-01501-t008:** Multi-class classification results using the original and reconstructed ECG datasets.

	Results Using the Original ECG Dataset				Results Using the Reconstructed ECG Dataset			
	Overall Precision	Overall Recall	F1 Score	Accuracy	Overall Precision	Overall Recall	F1 Score	Accuracy
VGG16	97.93%	98.08%	98.01%	98.03%	97.93%	98.08%	98.01%	98.03
VGG16 + SVM	97.93%	98.08%	98.01%	98.08%	97.93%	98.08%	98.01%	98.08%
InceptionNetv2	96.29%	95.79%	96.04%	96.08%	94.53%	92.22%	93.36%	93.14%
InceptionNetv2 + SVM	97.87%	98.25%	98.06%	95.79%	97.50%	96.67%	97.08%	96.67%
VGG19	99.42%	99.25%	99.34%	99.34%	97.93%	98.08%	98.01%	98.03%
VGG19 + SVM	99.42%	99.25%	99.34%	99.25%	97.33%	97.33%	97.33%	100%
ResNet50	98.85%	98.51%	98.68%	98.68%	96.70%	95.52%	96.11%	96.05%
ResNet50 + SVM	98.30%	97.76%	98.03%	97.76%	96.70%	95.52%	96.11%	95.52%
GoogleNet	96.16%	95.84%	96.00%	96.05%	95.27%	95.41%	95.34%	95.39%
GoogleNet + SVM	96.74%	96.58%	96.66%	96.58%	95.27%	95.41%	95.34%	95.41%

## Data Availability

The dataset used in this study is publicly available at https://data.mendeley.com/datasets/gwbz3fsgp8/1/ (accessed on 8 June 2025).

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
