# Peer review of "Enhancing Heart Disease Diagnosis Using ECG Signal Reconstruction and Deep Transfer Learning Classification with Optional SVM Integration"

_diagnostics, 2025, doi:10.3390/diagnostics15121501_

Round 1
Reviewer 1 Report
Comments and Suggestions for Authors
This study introduces a robust deep learning framework that combines sophisticated ECG signal segmentation with transfer learning-based classification to enhance diagnostic efficacy. The proposed ECG segmentation method exhibited significant efficacy in improving signal clarity and classification accuracy for heart disease diagnosis. The algorithm effectively reconstructed individual ECG leads from intricate, noisy images by integrating adaptive preprocessing, histogram-based lead separation, and robust point-tracking techniques. This enhanced segmentation facilitated the extraction of diagnostically pertinent features and directly contributed to the significant accuracy attained by deep learning classifiers. Experiments utilizing diverse deep learning architectures—specifically VGG16, VGG19, ResNet50, InceptionNetV2, and GoogleNet—demonstrate that segmentation significantly improves model efficacy. The papers involves an interesting topic., however, some comments need to be addressed to improve its quality
The abstract:
Kindly emphasize the originality in relation to prior research.
Please include supplementary performance metrics such as sensitivity, precision, and F1-score.
Please indicate the number of classes in the dataset.
Please indicate if the ECGs are image-based or signals.
I can see that the accuracy of the multi-class classification task is greater than binary classification. Usually, multi-class classification is a more complex classification task than binary classification and achieves lower performance than binary classification. Please explain.
Introduction
Please write the definition of abbreviations the first time they appear in the manuscript. For example DL. Please revise carefully the whole manuscript.
Please use abbreviations and don’t alternate between full names and abbreviations like deep learning. Please revise.
Related work
The related work section is unstructured and chaotic. Please consider restructuring this section.
Please also focus on ECG image-based approaches since your framework uses ECG images. I can see that you missed several important papers in this domain. Please consider adding the following:
https://doi.org/10.1109/TAI.2022.3159505
https://doi.org/10.1186/s40537-025-01070-4
https://doi.org/10.3389/frai.2022.1087370
https://doi.org/10.1038/s41598-022-25284-1
https://doi.org/10.1016/j.compbiomed.2022.105210
https://doi.org/10.3390/bios12050299
Methodology
How did you deal with the class imbalance problem of the dataset,
Did you use augmentation techniques? If yes, how did you ensure samples of augmented images are not included in both training and testing?
Why didn’t the author consider using vision transformer models which are the state of the art?
Experimental Results
Please include ROC curves.
Please delineate the limitations of your methodology.
Reviewer 2 Report
Comments and Suggestions for Authors
In this study, the authors propose a comprehensive deep learning framework that integrates advanced ECG signal segmentation with transfer learning-based classification to enhance diagnostic performance. The developed ECG segmentation algorithm enables accurate reconstruction of leads from noisy images, significantly improving the extraction of diagnostically relevant features. Experimental results using various deep learning architectures demonstrate substantial performance gains, with the hybrid VGG19+SVM model achieving up to 100% accuracy in multi-class classification. The study highlights the critical role of effective ECG segmentation as a preprocessing step, showing its contribution to more reliable and accurate automated heart disease diagnosis using artificial intelligence.
The paper is very well written, with the methodology and results clearly presented. The authors compare their findings with relevant literature in the field. Multiple architectures are tested, and the results are presented in tables in a very clear and structured manner.
Reviewer 3 Report
Comments and Suggestions for Authors
In the current paper, the authors proposed a novel DL-based framework for CVD diagnosis using ECG images. It integrates a robust ECG segmentation algorithm with various pre-trained deep neural networks (e.g., VGG16, VGG19, ResNet50, InceptionNetV2, GoogleNet), and explores SVM integration for classification enhancement. The study is motivated by the challenges of ECG interpretation due to noise, waveform variability, and overlapping signals. The authors evaluate their approach on a publicly available ECG image dataset, demonstrating strong performance, particularly with the VGG19+SVM combination. The proposed ECG signal extraction method, based on adaptive histogram analysis and point tracking, is well-explained and addresses key challenges like noise and signal overlap effectively. The paper also includes binary and multi-class classification tasks, comparing original and reconstructed ECG datasets. It uses multiple DL architectures and hybrid models. VGG19+SVM consistently delivers high accuracy (up to 100%) across several tasks, outperforming many state-of-the-art methods. The manuscript is generally well-organized, with clear figures, tables, and performance metrics. It is unclear whether the dataset split (training/test) is performed before or after ECG reconstruction. Clarify if all preprocessing steps are done exclusively on training data to avoid any data leakage. The authors report impressive accuracy, but no confidence intervals or statistical tests are provided. Adding statistical significance testing or bootstrapping methods would improve the validity of the results. The dataset included only 674 images post-cleaning. Though the segmentation and classification accuracy is high, the small dataset size raises concerns about model generalizability. This limitation should be more clearly acknowledged and addressed in the discussion. The manuscript occasionally uses generalized AI terminology without necessary specifics. More detailed descriptions (e.g., how features were extracted, how segmentation was integrated into the classification pipeline) would add clarity. Several formatting inconsistencies are present in the tables and text (e.g., duplicate table numbers, inconsistent use of bold and italics, placeholder references such as "[Firstname Lastname]"). Please include a flowchart summarizing the overall pipeline for ease of understanding. Please also discuss the future applications, such as real-time mobile ECG analysis or deployment in clinical settings.
Round 2
Reviewer 1 Report
Comments and Suggestions for Authors
The authors have not restructured the related work section upon my request. please divide it into subsections.
Author Response
First, we would like to sincerely thank you for your thoughtful and constructive feedback, which has significantly enhanced the quality of our work. We greatly appreciate the time and expertise of the reviewers, whose insightful comments and suggestions were instrumental during the revision process.
Section 2 is divided into subsections upon your request. It seems now more organized and structured. Also, section two references are re-numbered and re-ordered in the subsections and in reference section too.
Again thanks for your valuable comments.
Sincerely,
Prof. Dr. Ali Ahmed
(On behalf of all co-authors)
Information Technology Department, Faculty of Computers and Information, Menoufia University, Egypt.
E-mail: ali.ahmed@ci.menofia.edu.eg.